# Effects of *Priestia aryabhattai* on Phosphorus Fraction and Implications for Ecoremediating Cd-Contaminated Farmland with Plant–Microbe Technology

**DOI:** 10.3390/plants13020268

**Published:** 2024-01-17

**Authors:** Shenghan Yang, Yiru Ning, Hua Li, Yuen Zhu

**Affiliations:** 1Institute of Loess Plateau, Shanxi University, Taiyuan 030031, China; yshd163@163.com; 2School of Environment Science and Resources, Shanxi University, Taiyuan 030031, China; yrning01@163.com; 3Institute of Resources and Environment Engineering, Shanxi University, Taiyuan 030031, China; 4Shanxi Laboratory for Yellow River, Taiyuan 030031, China

**Keywords:** plant–microbe-combined technology, soil remediation, phosphorus fractions, *Priestia aryabhattai*, physical–chemical–microbial mechanism, ecoremediating Cd-contaminated farmland

## Abstract

The application of phosphate-solubilizing bacteria has been widely studied in remediating Cd-contaminated soil, but only a few studies have reported on the interaction of P and Cd as well as the microbiological mechanisms with phosphate-solubilizing bacteria in the soil because the activity of phosphate-solubilizing bacteria is easily inhibited by the toxicity of Cd. This paper investigates the phosphorus solubilization ability of *Priestia aryabhattai* domesticated under the stress of Cd, which was conducted in a soil experiment with the addition of Cd at different concentrations. The results show that the content of Ca_2_-P increased by 5.12–19.84%, and the content of labile organic phosphorus (LOP) increased by 3.03–8.42% after the addition of *Priestia aryabhattai* to the unsterilized soil. The content of available Cd decreased by 3.82% in the soil with heavy Cd contamination. *Priestia aryabhattai* has a certain resistance to Cd, and its relative abundance increased with the increased Cd concentration. The contents of Ca_2_-P and LOP in the soil had a strong positive correlation with the content of Olsen-P (*p* < 0.01), while the content of available Cd was negatively correlated with the contents of Olsen-P, Ca_2_-P, and LOP (*p* < 0.05). *Priestia aryabhattai* inhibits the transport of Cd, facilitates the conversion of low-activity P and insoluble P to Ca_2_-P and LOP in the soil, and increases the bioavailability and seasonal utilization of P in the soil, showing great potential in ecoremediating Cd-contaminated farmland soil with plant–microbe-combined technology.

## 1. Introduction

Cadmium is one of the most biotoxic heavy metals, and the pollution detection point over the standard rate of Cd is 7.0% in China’s farmland, which is at the top of the eight excess heavy metals [1,2]. Cadmium can be absorbed by plants, thereby impacting their growth and nutritional quality, and can generate serious threats to both the environment and human health through the food chain [3]. Farmland soil is identified as safe when the concentration of Cd is between 0.6 and 4 mg kg^−1^, which is considered to be the risk screening value and the risk control value (GB 15618-2018). Because the Cd-contaminated farmland may cause contamination of the agricultural food products, some measurements, such as agronomic control and alternative planting or other safety measures, have to be taken to reduce the risk and ensure the safe growth of agricultural crops [4,5]. The safe use of Cd-contaminated farmland in China has amounted to a relatively large area and is widely distributed [4]. The need for plant-based, microbe-based, and combined measures for the implementation of farmland eco-restoration to realize the safe utilization of this type of farmland has become pressing.

The application of phosphorus fertilizers is consistently at a high level in farmland due to its low seasonal availability [6,7,8]. However, the frequent and large-scale application of phosphorus fertilizers can cause the accumulation of Cd in agricultural soils, leading to environmental problems [9]. Therefore, increasing the utilization of soil P to reduce the input of low-quality phosphorus fertilizers is also an effective strategy by which to remediate Cd-contaminated farmland [10,11]. Phosphate-solubilizing bacteria (PSB) can convert insoluble phosphorus into soluble phosphorus that can be easily absorbed by plants and can increase the content of available phosphorus in the soil for the crop uptake [12]. Many kinds of microorganisms have the effect of phosphorus solubilization, of which *Priestia aryabhattai*, as a new type of microorganism, was first found at the top of the Earth’s stratosphere [13,14]. Previous studies have documented that *Priestia aryabhattai* is a powerful multi-stress-tolerant crop growth promoter which can improve plant productivity and enhance defense mechanisms against biotic and abiotic stresses [15,16]. The study of *Priestia aryabhattai* has concentrated recently on phosphorus solubilization in the soil. *Priestia aryabhattai* was found to increase the soil-available phosphorus and improve the growth and nutrition of mung bean and maize crops [17]. *Priestia aryabhattai* was also reported to secrete a large number of organic acids that can lead to phosphate solubility up to 388.62 g mL^−1^ [18]. A strain of *Priestia aryabhattai* was screened for the secretion of organic acids and extracellular phosphatases, which solubilized the insoluble phosphates in the soil and increased the content of available P [19]. However, the effect mechanisms of *Priestia aryabhattai* on the phosphorus fractions in Cd-contaminated farmland are yet to be revealed.

The interaction between phosphorus and cadmium elements is an effective means for phosphorus-solubilizing bacteria to remediate Cd-contaminated farmland [20]. Adhikari et al. demonstrated that *Enterobacter ludwigii* GAK2 effectively solubilized soil-insoluble phosphate and reduced the Cd content in Oryza sativa [21]. Wang et al. showed that phosphate-solubilizing bacteria promoted the release of available phosphorus, and P-Cd complex precipitation in the soil reduced the biological effectiveness of heavy metals, alleviating the toxic effect of cadmium on *Brassica juncea L.*, which proved that the combination of phosphate-solubilizing bacteria and plants is a restoration strategy with an appreciable potential to resolve Cd contamination [22]. Compared with exclusive phytoremediation and microbial remediation methods, plant–microbe-combined technology can improve reducing activity and overcome the challenges of low efficiency and a long bioremediation cycle [23]. The rational use of P can reduce the content of available Cd in soil, reduce the accumulation of Cd in plants, and improve plant resistance to the stress of Cd [9,24]. However, the interaction of P and Cd was also reported to increase the activity of Cd and the enrichment of Cd in plants [25]. Differences in P-Cd interaction may be related to a variety of factors, such as the soil type, soil condition, the form and level of P, and the changing activity of Cd under the corresponding conditions. The relationships are complicated for these reasons [4,26,27,28]. It remains unknown as to whether *Priestia aryabhattai* could be utilized as an efficient inoculant in phytoremediation systems for Cd-contaminated farmland. Therefore, it is particularly important to first study the effect of *Priestia aryabhattai* on the safe utilization of phosphorus fractions and phosphorus–cadmium interactions in Cd-contaminated agricultural soils.

In this study, we hypothesize that *Priestia aryabhattai* can influence phosphorus–cadmium interactions by increasing the bioavailability of phosphorus in the soil. To test our hypothesis, a strain of phosphate-solubilizing bacteria was isolated from agricultural soil and identified as *Priestia aryabhattai* in this study. The screened and domesticated *Priestia aryabhattai* was inoculated in the farmland soil with light, moderate, and heavy Cd contamination. The specific objectives of this study were to analyze the phosphorus solubility of *Priestia aryabhattai* and its interaction with P and Cd and crops by determining the content of soil inorganic and organic P, the content of available Cd, and the changes in bacterial abundance and community structure. The results support the innovative idea of developing and applying plant–*Priestia aryabhattai*-combined technology for ecoremediating Cd-contaminated farmland and increasing the utilization efficiency of phosphorus.

## 2. Results and Discussion

### 2.1. Strain Identification

A strain of phosphate-solubilizing bacteria Z3236-7 resistant to Cd was isolated in NBRIP medium based on halo formation (Figure 1a,b). The phylogenetic tree was constructed using MEGA 11 software, as shown in Figure 1c, and the phosphate-solubilizing bacteria were identified as *Priestia aryabhattai.*

### 2.2. Effect of Priestia aryabhattai on Available Phosphorus in Cd-Contaminated Soil

The available phosphorus (Olsen-P) is the phosphorus fraction of the soil which is accessible for plant uptake, and its content reflects the availability of phosphorus nutrients in the soil. The content of Olsen-P in the heavily Cd-contaminated soil (4.0 mg kg^−1^) was lower than that in the lightly and moderately Cd-contaminated soil (1.0 and 2.0 mg kg^−1^), as shown in Figure 2a. The reported results have shown that soluble P might react with Cd^2+^ to form a relatively stable Cd-containing phosphate. Cadmium (II) also could be directly adsorbed and immobilized by P-containing groups on the soil surface [29]. As compared with the treatment group without bacteria, the application of *Priestia aryabhattai* significantly increased the content of Olsen-P after 28 d (*p* < 0.05) (Figure 2b), indicating that *Priestia aryabhattai* could increase the content of soluble P in Cd-contaminated soil.

### 2.3. Effect of Priestia aryabhattai on the Inorganic Phosphorus Fractions in Cd-Contaminated Soil

The content of soil inorganic P forms under different treatments is shown in Table 1 and Table 2. All treatment groups with *Priestia aryabhattai* can significantly increase the content of Ca_2_-P in the soil (*p* < 0.05) as compared with the treatment group without exogenous bacteria (except N4-PSB). When compared with the non-sterilized soil treatment group, the soluble P in the sterilized soil was higher with the application of *Priestia aryabhattai*, which may be due to the great competitive result of *Priestia aryabhattai* over indigenous microorganisms [30,31]. The contents of Ca_8_-P, Ca_10_-P, Fe-P, and O-P have no significant improvement with *Priestia aryabhattai* from unsterilized soil (*p* > 0.05). The contents of Ca_8_-P, Ca_10_-P, Fe-P, and Al-P were decreased in sterilized soil with the addition of *Priestia aryabhattai*. The content of Ca_8_-P was decreased by 13.10% in S2-PSB and 8.02% in S4-PSB (*p* < 0.05). The content of Ca_10_-P was significantly decreased by 8.97% with S1-PSB treatments, and the content of Fe-P was decreased by 7.82% with S1-PSB and 6.25% with S2-PSB treatments (*p* < 0.05). The published results have shown that phosphate-solubilizing bacteria could promote the conversion of Ca_10_-P and Ca_8_-P to Ca_2_-P in Cd-Pb-contaminated soils by secreting oxalic acid, ascorbic acid, citric acid, and succinic acid [20]. Organic acids can also form chelates with Ca, Fe, and Al from insoluble inorganic phosphates to reduce the interaction of these metal ions with P and increase the availability of P [31,32]. O-P is a potential source of P that is sequestered by soil colloids and is rarely utilized by plants [30]. No significant difference was observed among the contents of O-P in all treatment groups. Inorganic P in the soil of each treatment group dominantly exists as Ca_10_-P, followed by Ca_8_-P, O-P, Fe-P, and Al-P, with Ca_2_-P being the least abundant, indicating that most of the accumulation of P was in the sustained releasing and inert forms [33]. The application of *Priestia aryabhattai* decreased the contents of Ca_8_-P, Ca_10_-P, Fe-P, and Al-P in the soil, significantly increased the contents of Ca_2_-P, but had no effect on O-P. The results suggest that *Priestia aryabhattai* increases the availability of P in the soil by promoting the transformation of inorganic P forms to Ca_2_-P. Qin et al. showed that phosphate-solubilizing bacteria could induce the precipitation of Cd(PO_3_)_2_ in the soil and increase the Ca_2_-P content in the plant rhizosphere, as well as promote the growth characteristics of plants, which provide bacterial resources and technical approaches to heavy metal pollution amelioration in farmland [20].

The transformation of inorganic P forms by *Priestia aryabhattai* was affected by Cd concentration. The content of Ca_2_-P decreased with the increasing Cd concentration after adding *Priestia aryabhattai.* The content of Ca_2_-P among all treatment groups at different Cd concentrations was 49.20, 45.25, 41.13, and 37.75 mg kg^−1^, respectively, in the sterilized soil, which were all significantly different (*p* < 0.05). But no significant difference was observed in the content of Ca_8_-P and O-P for all the different-Cd-concentration-treated groups. As compared with the treatment group with a Cd concentration of 0 mg kg^−1^, the application of *Priestia aryabhattai* in the sterilized soil significantly increased the contents of Ca_10_-P by 10.15% and Al-P by 9.36% in the group with 4.0 mg kg^−1^ of Cd treatment (*p* < 0.05). The content of Fe-P was significantly decreased by 6.00% (*p* < 0.05) in the group with 1.0 mg kg^−1^ of Cd treatment. The study indicated that *Priestia aryabhattai* mainly increased the availability of P by increasing the content of Ca_2_-P at different Cd concentrations. However, a certain inhibitory effect was reported with respect to phosphate solubilization under the stress of high-concentration Cd with the addition of *Priestia aryabhattai* [9,34].

### 2.4. Effect of Priestia aryabhattai on the Organic Phosphorus Fractions in Cd-Contaminated Soil

The contents of soil organic phosphorus fractions under each treatment are shown in Table 3 and Table 4. As compared with the treatment group without exogenous bacteria, the content of LOP in the soil increased with the application of *Priestia aryabhattai*, and the content of moderately labile organic phosphorus (MLOP) decreased, and no significant difference was observed for the non-sterilized soil treatment group. However, in sterilized soil, the content of LOP was significantly increased by 64.17%, 65.44%, 19.40%, and 19.30% at different Cd concentrations, respectively (*p* < 0.05), which was consistent with the reported results by Estrada et al. The addition of phosphate-solubilizing bacteria could increase the content of LOP [35]. The main components of LOP are phospholipids and nucleic acids, which can be uptaken by plants to meet their nutritional needs for growth [36]. With the application of *Priestia aryabhattai*, the contents of MLOP were significantly decreased by 16.21%, 17.50%, and 11.64% (*p* < 0.05) under treatments with Cd concentration < 4.0 mg kg^−1^ in the sterilized soil, respectively. In sterilized soil, *Priestia aryabhattai* significantly decreased the content of MROP under the Cd concentration of 0 and 1.0 mg kg^−1^ treatments and HROP under the Cd concentration of 1.0 mg kg^−1^ treatment (*p* < 0.05), and no significant difference for other treatments. This study indicated that *Priestia aryabhattai* facilitated the transformation of the stable organic P forms to the more-active organic P forms, which were further converted to available P. Other studies have shown that phosphate-solubilizing bacteria could mineralize organic P sources with phytase and acidic and alkaline phosphatases to increase the content of available P [37,38].

As compared with the treatment with Cd concentration at 0 mg kg^−1^, the application of *Priestia aryabhattai* in the sterilized soil significantly decreased the contents of LOP by 21.17% at Cd concentrations of 2.0 mg kg^−1^ and 22.29% at the Cd concentration of 4.0 mg kg^−1^ (*p* < 0.05). The contents of MLOP significantly increased by 9.18% and 17.30%, respectively (*p* < 0.05). The application of *Priestia aryabhattai* had no significant effect on MROP in the soil at different Cd concentrations. However, the contents of HROP in the sterilized soil significantly decreased by 11.70% and 12.87% under the treatments with Cd concentrations of 1.0 and 2.0 mg kg^−1^, respectively (*p* < 0.05). The results indicated that *Priestia aryabhattai* promoted the transformation of HROP to LOP in the lightly Cd-contaminated soil, and the high Cd concentration inhibited the mineralization of *Priestia aryabhattai.*

### 2.5. Relationship between Soil Inorganic Phosphorus, Organic Phosphorus Fractions and Available Cd

#### 2.5.1. Effect of *Priestia aryabhattai* on Available Cd Content in Soil

As shown in Figure 3, the application of *Priestia aryabhattai* increased the contents of available Cd by 12.69% for the unsterilized soil at the Cd concentration of 1.0 mg kg^−1^ compared to the treatment with the Cd concentration of 0 mg kg^−1^ (*p* < 0.05) While the content of available Cd at the Cd concentration of 2.0 mg kg^−1^ was not significantly different from that at the Cd concentration of 0 mg kg^−1^, the application of *Priestia aryabhattai* decreased the content of available Cd by 3.82% in the soil with 4.0 mg kg^−1^ of Cd contamination. The content of available Cd in the sterilized treatment group of 4.0 mg kg^−1^ Cd contamination was significantly lower than that of 0 mg kg^−1^, with a decrease of 16.34% (*p* < 0.05). The content of available Cd in the soil with 1.0 and 2.0 mg kg^−1^ of Cd contamination decreased by 18.16% and 7.88%, respectively, but there was no significant difference. Phosphate-solubilizing bacteria were reported as transforming a wide range of insoluble phosphates into soluble forms by secreting hydrogen ions, organic acids, etc., and the combination of PO_4_^3−^ with Cd can lead to the formation of phosphate salt [29]. Also, *Priestia aryabhattai* may reduce the mobility and bioavailability of Cd in the soil by biosorption and biological accumulation [20]. The study demonstrated that *Priestia aryabhattai* is a sustainable P-fertilizer which could dissociate the fixed P pool accumulated in soil and promote the growth characteristics of plants while reducing the bioavailability of cadmium. Therefore, *Priestia aryabhattai* combined with plants may be promising technology for the remediation of Cd-contaminated soil (Figure 4).

#### 2.5.2. Correlation Analysis between Soil Organic Phosphorus, Inorganic Phosphorus Fractions, and Available Cd

The correlation between the Olsen-P and the inorganic phosphorus and organic phosphorus fractions in Cd-contaminated soil after the application of *Priestia aryabhattai* is shown in Figure 5. Olsen-P has a great positive correlation with Ca_2_-P and LOP (*p* < 0.01), and a great negative correlation with Ca_8_-P, Ca_10_-P, and MLOP (*p* < 0.01). No significant correlation was observed between different inorganic P forms and organic P forms (except O-P and HROP). Ca_2_-P is negatively correlated with Ca_10_-P, Ca_8_-P, Al-P, and Fe-P (*p* < 0.01). Great positive correlation also exists between Ca_8_-P, Ca_10_-P, Al-P, and Fe-P (*p* < 0.01), indicating that *Priestia aryabhattai* increased the content of available P mainly by facilitating the transformation of inorganic P forms (except O-P) to Ca_2_-P and then increasing the content of available P. LOP has a great negative correlation with MLOP and MROP (*p* < 0.01), while a great positive correlation was observed between MLOP and MROP (*p* < 0.05), indicating that MLOP and MROP could increase available P from the transformation to LOP. Also, Ca_2_-P has a great positive correlation (*p* < 0.05) with LOP and great negative correlation (*p* < 0.01) with MLOP and MROP. No transformational relationship was observed between organic and inorganic P fractions. Organic P in the soil can be transformed into inorganic P by phosphatase enzymes produced by dephosphorylated bacteria [39,40].

The contents of Ca_2_-P and LOP have a negative correlation with the available Cd in the soil, and the content of LOP has a great negative correlation with the available Cd (*p* < 0.05). *Priestia aryabhattai* facilitated the conversion of stabilized P forms to active Ca_2_-P and LOP and released more PO_4_^3−^ to inhibit Cd migration, while Al-P, Fe-P, and MLOP show a significant positive correlation with the available Cd. A possible explanation may be that *Priestia aryabhattai* released organic acids and activated P along with Cd in the soil [41]. Therefore, *Priestia aryabhattai* is expected to be a remediation agent for Cd-contaminated soil.

### 2.6. Effect of Priestia aryabhattai on Microbial Diversity and Community Composition of Soils

#### 2.6.1. Soil Bacterial Alpha Diversity

The α diversity of soil microbial communities was indicated by four indicators: Chao1, Pielou evenness, Shannon, and Simpson. Chao1 reflects the abundance of the community. Pielou evenness characterizes the uniformity of the community. Shannon and Simpson reveal the abundance and uniformity of the community, which could reflect the diversity of the community. As shown in Table 5, the treatment of N4-PSB with the application of *Priestia aryabhattai* slightly decreased Chao1, Pielou evenness, Shannon, and Simpson as compared with the treatment of N4, which was consistent with the research results of Zhang et al. [9]. This may be attributed to the competition between indigenous and inoculated bacteria [42]. The reduced Cd tolerance of indigenous microorganisms in the severely Cd-contaminated soils caused the decreased activity, and the application of *Priestia aryabhattai* also resulted in a higher proportion than that in the soil with the <4.0 mg kg^−1^ Cd-contaminated group. The differences of the four indicators were not significant for the remaining treatments (*p* > 0.05).

#### 2.6.2. Changes in Bacterial Community Composition

Generally, the major bacteria in each treatment group include Proteobacteria, Actinobacteria Gemmatimonadota, Bacteroidota, Acidobacteriota, Chloroflexi, Firmicutes, and others, as shown in Figure 6a. Proteobacteria and Actinobacteria are considered to be the main strains of bacteria involved in soil nutrient recycling [43,44] and have the highest abundance in all the soil treatment groups. The average percentage of the two bacteria ranges from 52.36% to 64.11%. The relative abundance of Proteobacteria was reduced by 3.59% for N0-PSB, 8.50% for N1-PSB, 5.26% for N2-PSB, and 8.84% after the addition of *Priestia aryabhattai*. The reduced abundance of Proteobacteria was associated with the Cd stress and competitive effects of other microbial communities. Firmicutes are often found in extreme environments such as high-salt and heavy-metal-contaminated soil [45,46]. In the treatment groups without the addition of *Priestia aryabhattai*, the relative abundance of the Firmicutes was 0.63% (N0), 0.39% (N1), 0.41% (N2), and 0.35% (N4). The relative abundance increased to 3.75% (N0-PSB), 7.85% (N1-PSB), 8.29% (N2-PSB), and 12.69% (N4-PSB) in each group after the addition of *Priestia aryabhattai* and showed an increased trend with the increasing Cd concentration. This suggests that the addition of *Priestia aryabhattai* increased the resistance and adaptability of Firmicutes to the heavy metal stress.

The bacteria with a high relative abundance in each treatment group include *Sphingomonas*, *Lysobacter*, *Bacillus*, *Nocardioides*, *Gemmatimonasand* and others, as shown in Figure 6b. *Sphingomonas* had the highest relative abundance (except N4-PSB), and the average percentage ranges from 11.56% to 21.52%. The relative abundance of *Sphingomonas* decreased, and that of *Lysobacter* increased, with the addition of *Priestia aryabhattai*. The relative abundance of *Bacillus* substantially increased in all treatment groups and became the dominant bacteria in the community. The relative abundance of *Bacillus* in each group was 0.40% (N0), 0.20% (N1), 0.22% (N2), and 0.16% (N4) before the addition of *Priestia aryabhattai*. The relative abundance of *Bacillus* increased to 3.27% (N0-PSB), 6.64% (N1-PSB), 7.83% (N2-PSB), and 13.04% (N4-PSB) with the addition of *Priestia aryabhattai*. The relative abundance of *Bacillus* showed an increasing trend with the increasing Cd concentration. This suggested that *Priestia aryabhattai* had a high tolerance to the high concentration of Cd. *Nocardioides*, *Gemmatimonas*, and *Gitt-GS-136* also decreased with the addition of *Priestia aryabhattai*, which may be attributed to the competition between flora.

More studies have shown that phosphate-solubilizing bacteria could alter soil microbial community compositions and increase the connections between different soil microbes [37,47]. Inoculation with phosphate-solubilizing bacteria could increase the dominance of key phosphate-solubilizing bacteria in symbiotic patterns and the multi-functional potential of the core bacterial community, which indirectly increases the effectiveness of phosphorus in the soil and promotes plant growth [30,46]. The relationships between the relative abundance of the top 10 dominant bacteria and the soil P are shown in Figure 6c. Both *Bacillus* and *Lysobacter* are positively correlated with the contents of soil-available P, Ca_2_-P, and LOP. Many studies have reported that *Bacillus* has good resistance towards acid and heat as well as great environmental adaptability. It also has other functions such as P solubilization, plant growth, and probiotic potential promotion [30,48]. *Lysobacter* can produce lytic enzymes that disrupt cell walls and may facilitate the mineralization of organic P [30]. *Lysobacterwere* was reported to be positively correlated with the activity of soil AlP and the content of unstable organic P [49]. The inoculation of *Priestia aryabhattai* in this study increased the abundance of phosphate-solubilizing bacteria (*Lysobacter* and *Bacillus*), indicating that *Priestia aryabhattai* can increase the abundance of key bacteria in Cd-contaminated soils, that it has the potential to increase the available P, and that it indirectly affects the transformation of soil P forms.

## 3. Materials and Methods

### 3.1. Materials

*Priestia aryabhattai* was first screened from agricultural soil and domesticated under Cd stress conditions (50 mg L^−1^ Cd); then, it was further isolated and purified. The detailed procedure was as follows: A single colony of *Priestia aryabhattai* was spread on the National Botanical Research Institute’s phosphate growth medium (NBRIP) with the addition of 50 mg L^−1^ Cd (CdCl_2_·2.5H_2_O) then incubated at 37 °C with 180 rpm for 72 h [50]. The individual colony with visible halos was recurrently purified with NBRIP medium five times.

Soil was obtained from the 0–20 cm topsoil of farmland in Yuci District, Jinzhong, Shanxi, China (the crop grown on the farmland is wheat). The soil was naturally air-dried, with plant debris removed, and sieved through a 2 mm sieve. The aqueous solution of Cd chloride (CdCl_2_·2.5H_2_O) was prepared at three concentrations of 1.0, 2.0, and 4.0 mg kg^−1^ by following the guide for the safe utilization of light, moderate, and heavy Cd-contaminated soil. The soil was well-mixed with each Cd solution then left to equilibrate and age for 90 days [51].

### 3.2. Culture Medium

NBRIP inorganic phosphorus medium: The agar medium was made from 10.0 g of C_6_H_12_O_6_, 5.0 g of Ca_3_(PO_4_)_2_ or Hydroxyapatite, 5.0 g of MgCl_2_·6H_2_O, 0.25 g of MgSO_4_·7H_2_O, 0.20 g of KCl, 0.10 g of (NH_4_)_2_SO_4_, and 18 g–20 g of agar [52]. The pH of the medium ranges from 7.0 to 7.5. For the aqueous medium, 18–20 g of agar was replaced by 1000 mL of distilled water. The agar medium and aqueous medium were sterilized in an autoclave at 121 °C for 30 min.NBRIP organophosphorus medium: The agar medium was made from 10.0 g of C_6_H_12_O_6_, 5.0 g of calcium phytate, 5.0 g of MgCl_2_·6H_2_O, 0.25 g of MgSO_4_·7H_2_O, 0.20 g of KCl, 0.10 g of (NH_4_)_2_SO_4_, and 18–20 g of agar [52]. The pH of the medium ranges from 7.0 to 7.5. For the aqueous medium, 18–20 g of agar was replaced by 1000 mL of distilled water. The agar medium and aqueous medium were sterilized in an autoclave at 121 °C for 30 min.The seed growth medium was an LB medium (lysogeny broth), which was made from 10 g of peptone (biochemistry), 5.0 g of yeast plaster,10 g of NaCl, and 1000 mL of distilled water. The pH of the medium ranges from 7.2 to 7.4. The medium was sterilized in an autoclave at 121 °C for 20 min.

### 3.3. Soil Culture Experiment Design and Sample Collection

Indoor soil culture trials were conducted using a completely randomized block design. The test setup is shown in Table 6. Three treatments were included: inoculation of *Priestia aryabhattai* (N-PSB), soil sterilization (S), and inoculation of *Priestia aryabhattai* after soil sterilization (S-PSB). The unpasteurized soil served as the control group (N).

Sixteen treatments were conducted, and each treatment had three replicates. The pot for all treatments had 2.0 kg of soil, with a diameter of 16 cm and height of 10 cm. During the experiment, the soil water content in the pot was maintained at about 60–70% of the water holding capacity of the field soil. Soil samples were collected before incubation (0 d) and after 28 days’ incubation (28 d). The soil sample was separated to two portions, one of which was stored at −80 °C for the analysis of the soil microbial communities. The other portion was air-dried then sieved with a 2 mm sieve and then a 0.149 mm sieve for determining the contents of different phosphorus fractions and the available Cd.

### 3.4. Measurement Methods

The soil physical and chemical properties were determined with reference to soil agrochemical analysis [53]. The soil pH was determined using a pH meter (ST3100, OHAUS instrument (Changzhou) Co., Ltd., Changzhou, China) (soil:water = 1:2.5). Soil samples were digested at a high temperature with the concentrated sulfuric acid, and then the total nitrogen (TN) was determined using a fully automated Kjeldahl nitrogen analyzer (KD-310, Opsis, Sweden). The soil organic matter was determined via the potassium dichromate colorimetric method with a UV-Vis spectrophotometer (Cary 60 UV-vis, Agilent, USA). The total phosphorus and available phosphorus were determined via the HClO_4_-H_2_SO_4_ fusion Mo-Sb anti spectrophotometric method and the NaHCO_3_ solution Mo-Sb anti-spectrophotometric method with a UV-Vis spectrophotometer, respectively [54]. The available potassium was determined via the NH_4_OAc leaching-flame photometric method [55]. The total Cd was determined by an inductively coupled plasma mass spectrometer (ICP-MS, PerkinElmer, Waltham, MA, USA) with the soil first digested using HNO_3_-HClO_4_-HF (6:2:2) [56]. The available Cd was determined by the ICP-MS after the extraction with diethylenetriamine pentaacetic acid (DTPA) [57]. The soil physical and chemical properties are shown in Table 7.

The soil inorganic phosphorus was determined via the molybdenum antimony colorimetric method and classified with reference to the previous report by Gu et al [58]. The soil inorganic P fractions were sequentially extracted by adding extractants to 1.0 g of soil as follows: the extraction was conducted for Ca_2_-P by adding 50 mL of 0.25 M NaHCO_3_; Ca_8_-P by adding 50 mL of 0.50 M NH_4_OAc; Al-P by adding 50 mL of 0.50 M NH_4_F; Fe-P by adding 50 mL 0.10 M NaOH and 0.10 M Na_2_CO_3_; O-P by 40 mL 0.30 M sodium citrate, 1.0 g of sodium thiosulfate, and 10 mL of 0.50 M NaOH; and Ca_10_-P by adding 50 mL of 0.50 M H_2_SO_4_. All extractions were accomplished by vigorously shaking the mixture then centrifuging it and taking the supernatant for the assay. 

The soil organic P was determined via the Mo-Sb anti spectrophotometric method and classified with reference to the modified Bowman–Cole method of Fan et al. [59,60]. The soil organic phosphorus fractions were sequentially extracted by adding extractant to 1.0 g of soil as follows: the most-active organophosphorus was extracted by adding 20 mL of 0.25 M NaHCO_3_; the moderately active organophosphorus was extracted by adding 50 mL of 1.0 M H_2_SO_4_; and the moderately stabilized and highly stabilized organophosphorus were extracted by adding 50 mL of 0.050 M NaOH and adjusting the pH of soil solution to 3; then, they were determined separately.

### 3.5. Analysis of Microbial Community Structure

The isolation and characterization of Cd-resistant *Rhizopus solani* were conducted using the 16s rDNA gene sequencing method with primers 27F and 1492R for PCR amplification. DNA sequencing was performed by Shanghai Parsonage Biotechnology and subjected to BLAST comparison in the NCBI GenBank database.

### 3.6. Statistical Analysis of Data

The experimental data were analyzed via one-way ANOVA using SPSS 24.0 (the significance level was set at *p* < 0.05), and the results were expressed as “mean ± SE”. The graphs were prepared using Origin 2021. The biological analysis of the microbial community structure and function, as well as its graphical visualization, were achieved by using IBM SPSS Statistics 24 and Origin 2021. The redundancy analysis was conducted using Canoco 5.0 software with soil inorganic P, organic P forms, and available Cd as explanatory variables and the levels of the soil bacterial community genus as the species variables. 

## 4. Conclusions

In summary, the current paper took *Priestia aryabhattai* as an example and investigated the chemical and microbial mechanism for the first time, providing a strong basis for developing plant–microbe-combined remediation technology. *Priestia aryabhattai* can significantly increase the content of available P in the soil and decrease the content of available Cd by 3.82% in severely Cd-contaminated soil. *Priestia aryabhattai* significantly affected the contents of inorganic P and organic P in the soil and their transformation, and it facilitated the conversion of Ca_8_-P, Al-P, Fe-P, and Ca_10_-P into Ca_2_-P and the conversion of MLOP and MROP into LOP, thus improving the availability of soil Olsen-P components and P. The relative abundance of soil *Bacillus* increased after the inoculation of *Priestia aryabhattai* and also increased with the increasing Cd concentration. *Bacillus* also had a high tolerance to the high concentration of Cd. *Priestia aryabhattai* can indirectly affect the transformation of soil P forms by affecting the structure of the bacterial community. 

This study is the first to provide a systematic chemical mechanism for the plant–microbe-combined technology used to ecoremediate Cd-contaminated farmland. Focusing on the study of phosphorus fertilizers in cadmium-contaminated farmland, the transformation and availability of phosphorus fractions in soil under the application of PSB were analyzed in detail, which provided a solid theoretical foundation for the remediation of Cd-contaminated farmland via plant–microbe-combined technology, and also presented a new solution to increase the seasonal availability of phosphorus fertilizer.

However, the interaction between plants and PSB is still unclear in plant rhizosphere soil. Future studies can be further combined with the determination of P-related enzymatic activities and multi-omics (plants, microbes, phenotype, transcriptomics) analysis to reveal the signal exchange and molecular interactions between plant, microbe, and soil phosphorus.

## Figures and Tables

**Figure 1 plants-13-00268-f001:**
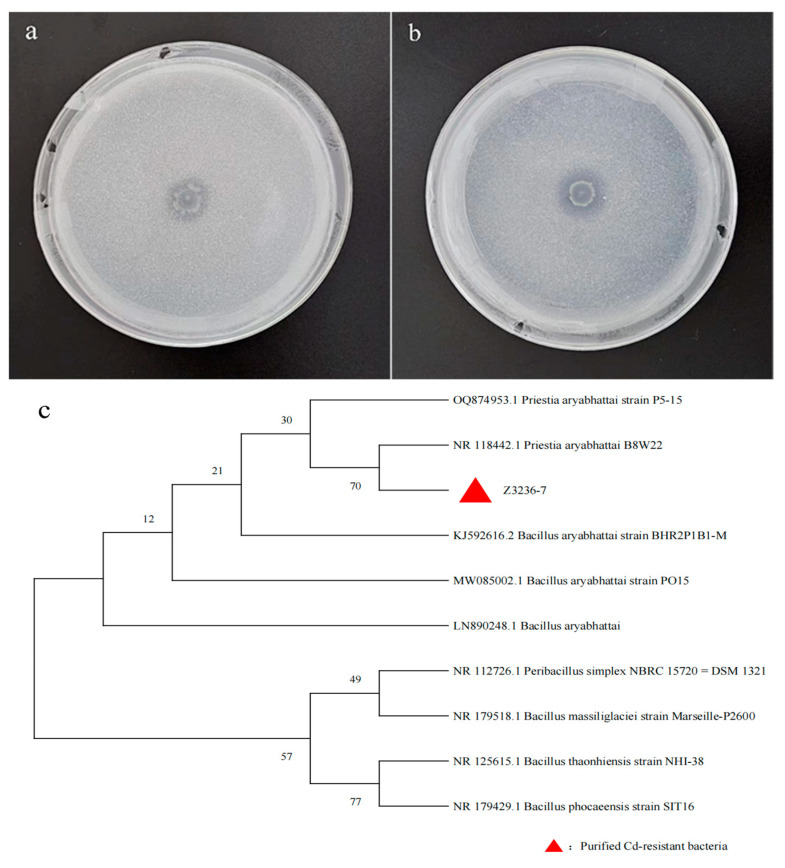
Purified Cd-resistant phosphate-solubilizing bacteria (**a**) as inorganic phosphorus medium and (**b**) as organophosphorus medium, and (**c**) phylogenetic tree of phosphate-solubilizing bacteria based on 16S rDNA sequence.

**Figure 2 plants-13-00268-f002:**
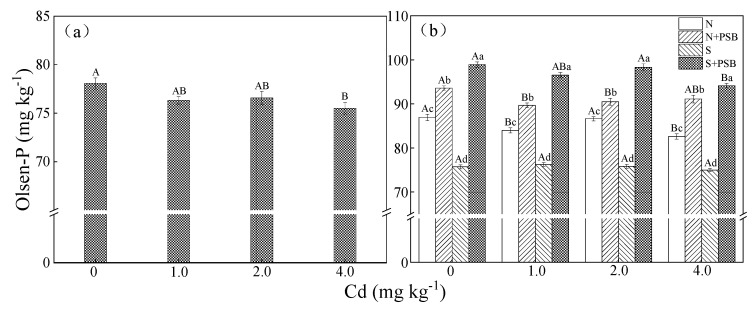
The content of Olsen-P in the soil of different treatment groups at 0 d (**a**) and 28 d (**b**). Note: error bars indicate standard error of the means (n = 3), different capital letters indicate significant differences between different Cd concentration at the same treatment (*p* < 0.05), and different lower-case letters indicate significant differences between different treatments at the same Cd concentration (*p* < 0.05).

**Figure 3 plants-13-00268-f003:**
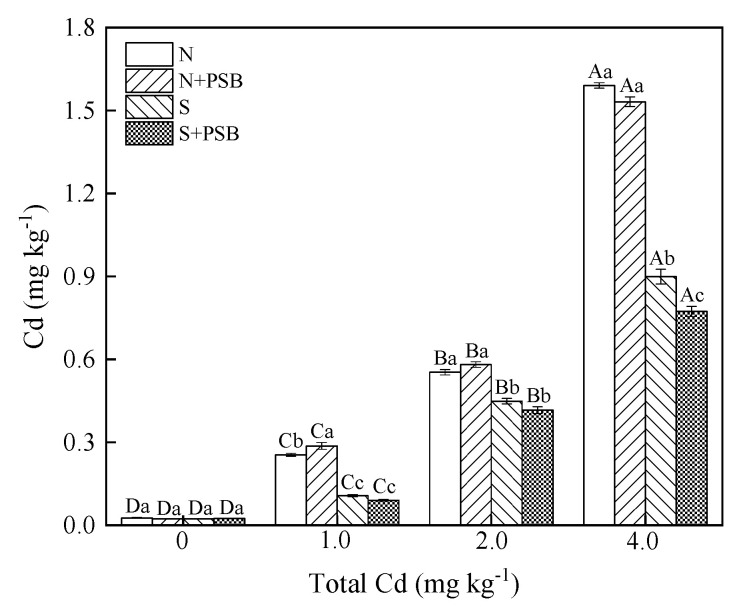
The content of available Cd in the soil in different treatment groups. Note: Error bars indicate standard error of the means (n = 3). Different capital letters indicate significant differences between different Cd concentrations at the same treatment (*p* < 0.05), and different lowercase letters indicate significant differences between different treatments at the same Cd concentration (*p* < 0.05).

**Figure 4 plants-13-00268-f004:**
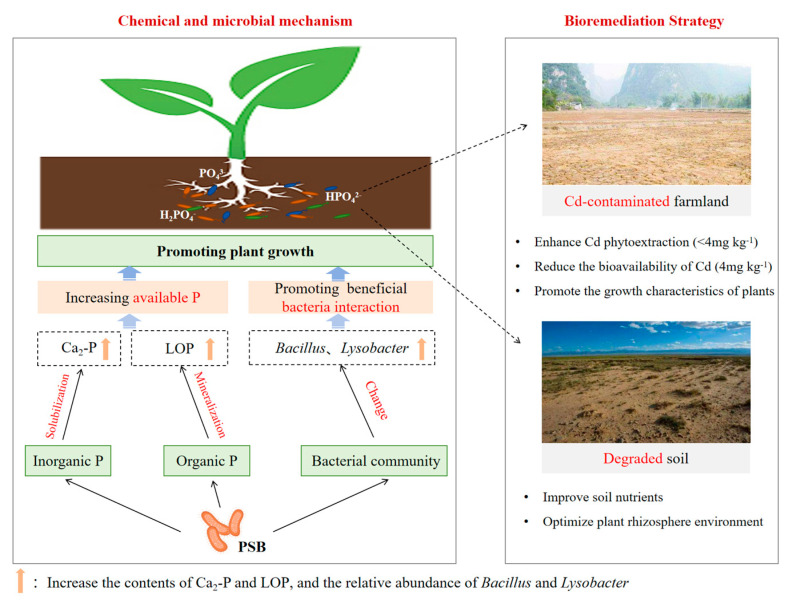
Schematic diagram of plant–microbe-combined technology ecoremediating Cd-contaminated farmland.

**Figure 5 plants-13-00268-f005:**
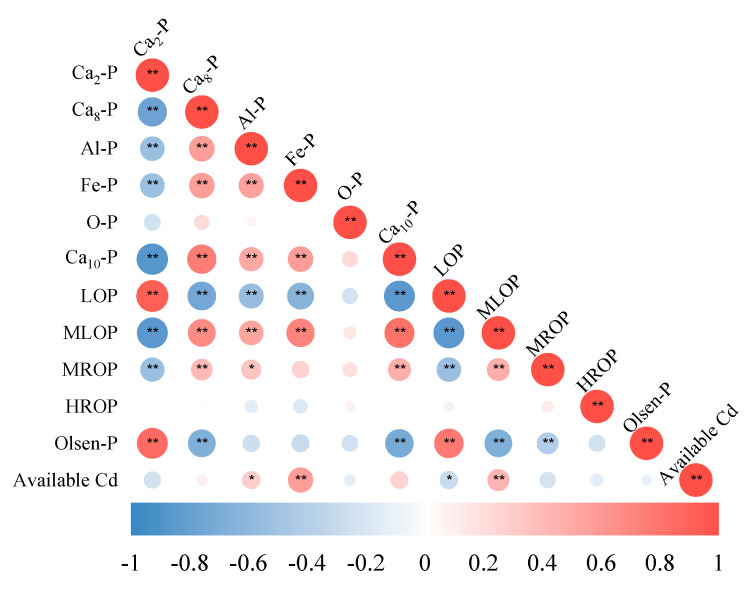
Correlation analysis of inorganic P, organic P forms, and available Cd of Cd-contaminated soil (28 d) after the application of *Priestia aryabhattai.* Note: red and blue circles indicate positive and negative correlation, respectively. Circles, from big to small indicate, indicate the decreased correlation from high to low. “*” sign in the circle indicates a significant difference between indicators (*p* < 0.05); “**” indicates a significant difference (*p* < 0.01) (Pearson correlation analysis).

**Figure 6 plants-13-00268-f006:**
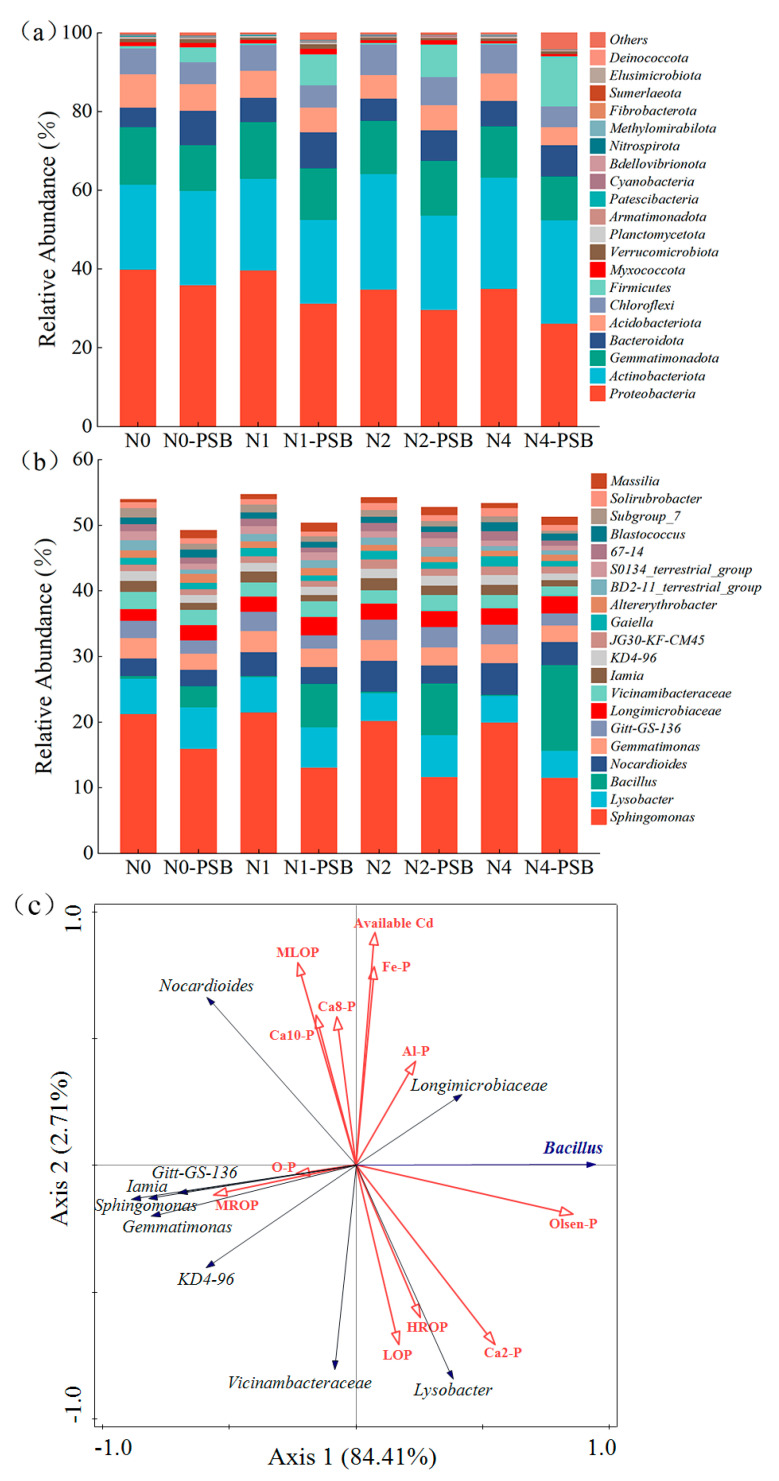
Correlation analysis of soil microbial community composition with environmental factors. (**a**) Relative abundance of soil-dominant bacterial phyla (top 20). (**b**) Relative abundance of dominant soil bacterial genera (top 20). (**c**) Redundancy analysis (RDA) of soil-dominant bacterial genera and soil key phosphorus fractions: Cd in the active state (Note: the red arrows indicate the available Cd and phosphorus fraction, and the blue arrow indicate *Bacillus*).

**Table 1 plants-13-00268-t001:** Various inorganic phosphorus fractions in the initial soil.

Inorganic Phosphorus Fractions	Ca_2_-P	Ca_8_-P	Al-P	Fe-P	O-P	Ca_10_-P
phosphorus content (mg kg^−1^)	31.03 ± 0.86	127.36 ± 3.15	19.68 ± 0.28	32.59 ± 0.66	116.67 ± 1.91	323.75 ± 4.45

**Table 2 plants-13-00268-t002:** Various inorganic phosphorus fractions under different treatments.

Total Cd (mg kg^−1^)	Treatment	Ca_2_-P	Ca_8_-P	Al-P	Fe-P	O-P	Ca_10_-P
0	N0	40.72 ± 0.83 Ab	114.64 ± 2.44 Bc	18.12 ± 0.50 Aa	26.54 ± 0.46 Cb	112.99 ± 1.98 Aa	297.18 ± 4.44 Aa
	N0-PSB	48.80 ± 1.01 Aa	110.04 ± 3.04 Ac	15.90 ± 0.39 Ab	27.38 ± 0.42 Bb	112.50 ± 1.43 Aa	287.76 ± 5.19 Ba
	S0	30.40 ± 1.33 Ac	126.48 ± 3.63 Aa	19.05 ± 0.39 Aa	30.85 ± 0.51 Ba	113.92 ± 1.26 Aa	324.23 ± 4.98 Aa
	S0-PSB	49.20 ± 0.68 Aa	119.85 ± 2.55 Aab	18.16 ± 0.31 Ba	29.99 ± 0.41 Ba	112.76 ± 2.07 a	285.59 ± 5.53 Ba
1.0	N1	41.00 ± 0.68 Ab	115.25 ± 1.79 Bb	16.49 ± 0.41 Bb	27.17 ± 0.58 Cb	113.02 ± 1.49 Aa	296.95 ± 5.92 Ab
	N1-PSB	45.99 ± 0.89 Ba	116.20 ± 2.66 Ab	18.84 ± 0.31 Ba	27.32 ± 0.54 Bb	112.48 ± 1.92 Aa	291.29 ± 7.06 ABb
	S1	29.93 ± 1.09 Ac	132.56 ± 2.05 Aa	19.63 ± 0.43 Aa	30.58 ± 0.44 Ba	113.79 ± 2.11 Aa	326.19 ± 5.69 Aa
	S1-PSB	45.25 ± 0.73 Ba	118.79 ± 2.60 Aa	18.73 ± 0.47 ABa	28.19 ± 0.45 Ab	112.63 ± 1.84 Aa	296.93 ± 5.25 Bb
2.0	N2	39.43 ± 0.43 Ab	117.48 ± 0.86 ABb	18.89 ± 0.37 Aa	30.15 ± 0.46 Bb	112.95 ± 1.38 Aa	297.67 ± 5.74 Aa
	N2-PSB	44.52 ± 0.82 Ba	118.00 ± 3.55 Ab	18.80 ± 0.49 Ba	30.72 ± 0.67 Ab	112.03 ± 1.87 Aa	298.55 ± 6.66 ABa
	S2	28.85 ± 0.92 Ac	133.51 ± 3.36 Aa	19.32 ± 0.37 Aa	32.63 ± 0.39 Aa	113.64 ± 2.09 Aa	325.62 ± 5.60 Aa
	S2-PSB	41.13 ± 0.77 Cb	116.02 ± 2.79 Ab	18.34 ± 0.51 Ba	30.59 ± 0.49 Bb	112.76 ± 1.61 Aa	301.60 ± 6.89 ABa
4.0	N4	36.93 ± 0.84 Ba	121.81 ± 2.46 Aab	18.43 ± 0.43 Aa	32.39 ± 0.68 Aa	112.50 ± 0.85 Aa	311.54 ± 4.69 Aa
	N4-PSB	38.82 ± 0.42 Ca	119.53 ± 3.04 Ab	19.13 ± 0.42 Ba	31.39 ± 0.61 Aab	111.75 ± 1.21 Aa	307.41 ± 4.90 Aa
	S4	28.18 ± 1.03 Ab	128.87 ± 3.56 Aa	19.40 ± 0.55 Aa	30.34 ± 0.51 Bb	113.83 ± 1.62 Aa	327.90 ± 6.32 Aa
	S4-PSB	37.75 ± 0.78 Da	118.54 ± 2.64 Ab	19.86 ± 0.40 Aa	30.97 ± 0.45 Bab	112.80 ± 1.75 Aa	314.58 ± 4.31 Aab

Note: Error bars indicate standard error of the mean (n = 3). Different capital letters indicate significant differences between different Cd concentrations at the same treatment (*p* < 0.05), and different lowercase letters indicate significant differences between different treatments at the same Cd concentration (*p* < 0.05).

**Table 3 plants-13-00268-t003:** Various organic phosphorus fractions in the initial soil.

Organic Phosphorus Fractions	LOP	MLOP	MROP	HROP
Phosphorus content (mg kg^−1^)	22.43 ± 0.35	101.28 ± 1.28	18.44 ± 0.51	22.34 ± 0.32

LOP: labile organic phosphorus; MLOP: moderately labile organic phosphorus; MROP: moderately resistant organic phosphorus; HROP: highly resistant organic phosphorus.

**Table 4 plants-13-00268-t004:** Various organic phosphorus fractions under different treatments.

Total Cd (mg kg^−1^)	Treatment	LOP	MLOP	MROP	HROP
0	N0	32.30 ± 0.87 Ac	86.43 ± 1.81 Bb	17.04 ± 0.66 Ab	22.34 ± 0.47 Bb
	N0-PSB	35.02 ± 1.18 Ab	87.22 ± 1.88 Bb	16.59 ± 0.67 Ab	24.07 ± 0.54 Aa
	S0	22.88 ± 0.56 ABd	103.98 ± 1.14 Aa	19.85 ± 0.32 Aa	22.69 ± 0.42 Bb
	S0-PSB	37.55 ± 0.53 Aa	87.12 ± 2.13 Bb	17.29 ± 0.86 Ab	23.00 ± 0.32 Aab
1.0	N1	33.56 ± 1.48 Aa	89.52 ± 2.12 Bb	17.22 ± 0.65 Ab	23.93 ± 0.46 Aab
	N1-PSB	34.69 ± 1.14 Aa	88.43 ± 3.13 Bb	17.35 ± 0.39 Ab	23.01 ± 0.72 ABb
	S1	21.79 ± 1.23 Bb	104.43 ± 4.17 Aa	19.16 ± 0.51 ABa	24.89 ± 0.46 Aa
	S1-PSB	36.05 ± 1.77 Aa	86.15 ± 1.51 Bb	16.79 ± 0.75 Ab	20.31 ± 0.42 Bc
2.0	N2	28.74 ± 0.52 Ba	98.48 ± 2.03 Ab	18.28 ± 0.47 Aa	22.21 ± 0.46 Bb
	N2-PSB	29.61 ± 0.57 Ba	92.90 ± 3.31 ABb	17.11 ± 0.62 Aa	24.38 ± 0.47 Aa
	S2	24.79 ± 0.51 Ab	107.65 ± 3.80 Aa	17.67 ± 0.46 Ba	21.11 ± 0.58 Cbc
	S2-PSB	29.60 ± 0.87 Ba	95.12 ± 3.15 Ab	17.92 ± 0.67 Aa	20.04 ± 0.27 Bc
4.0	N4	27.53 ± 0.51 Ba	103.12 ± 2.66 Aa	17.55 ± 0.33 Aa	22.16 ± 0.57 Ba
	N4-PSB	28.77 ± 0.34 Ba	97.16 ± 1.98 Aa	15.74 ± 0.59 Ab	22.09 ± 0.53 Bb
	S4	24.46 ± 0.76 Ab	106.28 ± 3.92 Aa	17.76 ± 0.64 Ba	23.05 ± 0.46 Ba
	S4-PSB	29.18 ± 0.58 Ba	102.19 ± 3.77 Aa	18.47 ± 0.64 Aa	22.73 ± 0.66 Aa

Note: Error bars indicate standard error of the means (n = 3). Different capital letters indicate significant differences between different Cd concentrations at the same treatment (*p* < 0.05), and different lowercase letters indicate significant differences between different treatments at the same Cd concentration (*p* < 0.05).

**Table 5 plants-13-00268-t005:** Soil bacterial α-diversity index under different treatments.

Total Cd (mg kg^−1^)	Treatment	Chao1	Pielou Evenness	Shannon	Simpson
0	N0	3710.53 ± 147.37 a	0.8695 ± 0.0011 a	10.22 ± 0.04 a	0.99773 ± 0.00005 a
	N0-PSB	3367.33 ± 111.69 a	0.8648 ± 0.0011 a	10.05 ± 0.02 a	0.99722 ± 0.00009 a
1.0	N1	3394.54 ± 94.37 a	0.8651 ± 0.0003 a	10.06 ± 0.03 a	0.99748 ± 0.00004 a
	N1-PSB	3672.31 ± 74.51 a	0.8594 ± 0.0029 a	10.08 ± 0.02 a	0.99705 ± 0.00027 a
2.0	N2	3607.65 ± 97.56 a	0.8704 ± 0.0015 a	10.19 ± 0.05 a	0.99781 ± 0.00009 a
	N2-PSB	3748.70 ± 25.88 a	0.8627 ± 0.0045 a	10.13 ± 0.05 a	0.99719 ± 0.00056 a
4.0	N4	3817.68 ± 61.15 a	0.8733 ± 0.0027 a	10.28 ± 0.03 a	0.99792 ± 0.00007 a
	N4-PSB	3425.09 ± 193.32 a	0.8206 ± 0.0227 b	9.54 ± 0.32 b	0.98690 ± 0.00580 b

Note: Data are means ± standard error in three independent treatments. Different letters within a column indicate a significant difference at *p* < 0.05.

**Table 6 plants-13-00268-t006:** Experimental treatment group.

	Total Cd (mg kg^−1^)	0	1.0	2.0	4.0
processing conditions	unpasteurized soil	N0	N1	N2	N4
	unpasteurized soil + PSB	N0-PSB	N1-PSB	N2-PSB	N4-PSB
	sterilized soil	S0	S1	S2	S4
	sterilized soil + PSB	S0-PSB	S1-PSB	S2-PSB	S4-PSB

*Priestia aryabhattai* is represented by PSB.

**Table 7 plants-13-00268-t007:** The soil physical and chemical properties.

Projects	Numerical Value
pH	8.67 ± 0.05
Total nitrogen (g kg^−1^)	1.324 ± 0.082
Organic matter (g kg^−1^)	6.427 ± 0.071
Total phosphorus (mg kg^−1^)	715.761 ± 8.930
Available phosphorus (mg kg^−1^)	76.828 ± 1.399
Available potassium (mg kg^−1^)	149.953 ± 5.648
Total Cd (mg kg^−1^)	0.056 ± 0.003
Available Cd (mg kg^−1^)	0.027 ± 0.003

## Data Availability

The data presented in this study are available in the graphs and tables provided in the manuscript.

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
