# Peer review of "Effects of Priestia aryabhattai on Phosphorus Fraction and Implications for Ecoremediating Cd-Contaminated Farmland with Plant–Microbe Technology"

_plants, 2024, doi:10.3390/plants13020268_

Round 1

Reviewer 1 Report

Comments and Suggestions for Authors

This paper holds significant research importance in the field of plant ecological restoration. The authors conducted a well-designed experiment, and several phenomena presented in the Plant-microbe combined technology are intriguing. The results demonstrate promising prospects for the ecological restoration of heavy metal-polluted farmland. Papers with minor revisions may be accepted for publication.

Detailed comments are as follows:

1. It is recommended to use italic text for the Latin names of bacteria in the article, such as Priestia aryabhattai mentioned in the keyword section, Enterobacter ludwigii GAK2, and Brassica juncea L mentioned in the third paragraph of the introduction.

2. The title of the article is "Cd-contaminated farmland," and the keyword is "Cd-polluted farmland." It is suggested that the author unify the expression of the term.

3. The title of the article is "phosphorus fractions," while the keywords, Table 3, and Table 5 repeatedly express "phosphorus forms." It is suggested that they be uniformly expressed as "phosphorus fractions."

Comments on the Quality of English Language

Minor editing of English language required

Author Response

Comments 1:

It is recommended to use italic text for the Latin names of bacteria in the article, such as Priestia aryabhattai mentioned in the keyword section, Enterobacter ludwigii GAK2, and Brassica juncea L mentioned in the third paragraph of the introduction.

Response 1:

Thank you for pointing this out. We agree with this comment. Therefore, we have standardized the formatting of Latin names of bacteria in manuscripts to italics as bellow:

Priestia aryabhattai (see page 1, lines 29-30)

Adhikari et al. demonstrated that Enterobacter ludwigii GAK2 effectively solubilized soil insoluble phosphate and reduced the Cd content in Oryza sativa [21]. (see page 2, lines 67-69)

Wang et al. showed that phosphate solubilizing bacteria promoted the release of available phosphorus, and P-Cd complex precipitation in the soil made the biological effectiveness of the heavy metals reduce, alleviating the toxic effect of cadmium on Brassica juncea L, which ...... (see page 2, line 72)

Comments 2:

The title of the article is "Cd-contaminated farmland," and the keyword is "Cd-polluted farmland." It is suggested that the author unify the expression of the term.

Response 2:

Thank you for your careful review of our manuscript. We have unified the expression as "Cd-contaminated farmland" in the article as follow:

Eco-remediating Cd-contaminated farmland (see page 1, line 30)

Comments 3:

The title of the article is "phosphorus fractions," while the keywords, Table 3, and Table 5 repeatedly express "phosphorus forms." It is suggested that they be uniformly expressed as "phosphorus fractions."

Response 3:

Thank you for your careful review of our manuscript. We have unified the expression as "phosphorus fractions" in the article as follows:

Phosphorus fractions (see page 1, line 29)

The other portion was air-dried, sieved with 2 mm sieve and then 0.149 mm sieve for determining the contents of different phosphorus fractions and the available Cd. (page 4, line 139)

Inorganic phosphorus fractions (page 9, line 252)

Organic phosphorus fractions (page 10, line 290)

Reviewer 2 Report

Comments and Suggestions for Authors

In this study, the interaction of P and Cd as well as the microbiological mechanisms of a newly isolated Cd tolerant bacteria Priestia aryabhattai was explored. It was found that Priestia aryabhattai can significantly increase the content of available P in the soil, and decrease the content of available Cd by 3.82% in the severely Cd-contaminated soil. Thus, the application of plant-microbial combined remediation technology is of great importance for the restoration of heavy metal polluted farmland. The study is innovative and well designed, meanwhile the manuscript is well written. The data provided are informative. This paper is of great significance in the design of plant-microbial combined remediation technology. The manuscript can be accepted for publication after minor revision.

Some detailed comments for revision are as follows:

1. In the last line of the first paragraph of Part 3.4, reference label [45,46] in blue font.

2. In the introduction, cadmium-contaminated soil is proposed to be unified into Cd-contaminated farmland.

3. The significant numbers of data in this study need to be unified, such as table 2.

4. In Part 2.2, the formulation of the medium is recommended to provide the relevant references.

5. In part 3.1, the phylogenetic tree in Figure 1 is not labeled with the letter (c), which is suggested to be added for clarity by the authors.

6. In Part 3.6.2 and Figure 6(a), “N+PSB" is suggested to be unified as "N-PSB".

Author Response

Thank you very much for taking time out of your busy schedule to give us constructive suggestions, which are very helpful for us to improve our paper. We have made correction according to the comments point by point, which we hope meet with approval.

Comments 1:

In the last line of the first paragraph of Part 3.4, reference label [45,46] in blue font.

Response 1:

Thank you for pointing this out. We have changed the font of [45,46] to blue as follow:

Other studies have shown that phosphate solubilizing bacteria could mineralize organic P sources with phytase, acidic and alkaline phosphatases, to increase the content of available P [48,49] (see page 10, line 279)

Comments 2:

In the introduction, cadmium-contaminated soil is proposed to be unified into Cd-contaminated farmland.

Response 2:

Thank you for your careful review of our manuscript. We have unified the expression as "Cd-contaminated farmland" in the article as follows:

Because the Cd-contaminated farmland may cause contamination for the agricultural food products. (see page 1, line 39)

It remains unknown whether Priestia aryabhattai could be utilized as an efficient inoculant in phyto-remediation systems for Cd-contaminated farmland. (see page 3, line 83)

Comments 3:

The significant numbers of data in this study need to be unified, such as table 2.

Response 3:

Thank you for your careful review of our manuscript. We have unified the format of letters of significant difference in manuscript as follow:

The contents of Ca2-P and LOP in soil were have a great positive correlation with the content of Olsen-P (P < 0.01), while the content of available Cd was negatively correlated with the contents of Olsen-P, Ca2-P and LOP (P < 0.05). (see page 1, lines 22-24)

Comments 4:

In Part 2.2, the formulation of the medium is recommended to provide the relevant references.

Response 4:

Thank you for pointing this out. We have supplemented the references for media formulations as follows:

NBRIP Inorganic Phosphorus Medium. The agar medium was made from 10.0 g of C6H12O6, 5.0 g of Ca3(PO4)2 or Hydroxyapatite, 5.0 g of MgCl2·6H2O, 0.25 g of MgSO4·7H2O, 0.20 g of KCl, 0.10 g of (NH4)2SO4 and 18 g-20 g of agar [31]. (see page 3, line 115)

NBRIP Organophosphorus Medium. The agar medium was made from 10.0 g of C6H12O6, 5.0 g of calcium phytate, 5.0 g of MgCl2·6H2O, 0.25 g of MgSO4·7H2O, 0.20 g of KCl, 0.10 g of (NH4)2SO4, and 18 g-20 g of agar [31]. (see page 4, line 120)

Reference

Nautiyal, C.S. An efficient microbiological growth medium for screening phosphate solubilizing microorganisms. Fems Microbiology Letters. 1999, 170, 265-270. [https://doi.org/10.1111/j.1574-6968.1999.tb13383.x]

Comments 5:

In part 3.1, the phylogenetic tree in Figure 1 is not labeled with the letter (c), which is suggested to be added for clarity by the authors.

Response 5:

Thank you for your helpful suggestion. We have already added the letter (c) in Figure 1 as follow:

Comments 6:

In Part 3.6.2 and Figure 6(a), “N+PSB" is suggested to be unified as "N-PSB".

Response 6:

Thank you for pointing this out. We have unified "N+PSB" in Section 3.6.2 and Figure 6 into" N-PSB " as follows:

The average percentage of the two bacteria ranges from 52.36% to 64.11%. The relative abundance of Proteobacteria was reduced by 3.59% for N0-PSB, 8.50% for N1-PSB, 5.26% for N2-PSB, and 8.84% after the addition of Priestia aryabhattai. (see page 15, lines 378-381)

The relative abundance increased to 3.75% (N0-PSB), 7.85% (N1-PSB), 8.29% (N2-PSB), and 12.69% (N4-PSB) in each group after the addition of Priestia aryabhattai, and showed an increased trend with the increasing Cd concentration. (see page 15, lines 384-388)

Sphingomonas had the highest relative abundance (except N4-PSB), and the average percentage ranges from 11.56% to 21.52%. (see page 15, line 392)

The relative abundance of Bacillus increased to 3.27% (N0-PSB), 6.64% (N1-PSB), 7.83% (N2-PSB), and 13.04% (N4-PSB) with the addition of Priestia aryabhattai. (see page 16, line 398)

Figure 6. Correlation analysis of soil microbial community composition with environmental factors.

4. Response to Comments on the Quality of English Language

Reviewer 3 Report

Comments and Suggestions for Authors

This is a research article that explores the relationship between soil phosphorus, cadmium (Cd) concentration, and the use of Priestia aryabhattai to remediate Cd-contaminated soil. The study found that the application of Priestia aryabhattai increased the content of available Cd in soil and the content of Ca2-P in the soil. The authors suggest that Priestia aryabhattai could be utilized as an efficient inoculant in phyto-remediation systems for Cd-contaminated soil. The article also discusses the importance of safe utilization of Cd-contaminated farmland and the need for plant-based and microbe-based measures for farmland Eco-restoration.

The strengths of the manuscript could be seen in its focus on a relevant and important environmental issue - heavy metal contamination in agricultural soils. The research seems to be well-referenced, citing numerous other studies, which suggests a thorough review of existing literature. The authors also declare no conflict of interest and the research received no external funding, which could add to the credibility of the study.

The manuscript also seems to propose a solution to the problem, which is the use of phosphate-solubilizing bacteria to increase the bioavailability and seasonal utilization of phosphorus in the soil, thereby aiding in the eco-remediation of cadmium-contaminated farmland soil. This could be seen as a significant applicative value of the study.

The introduction clearly states the purpose of the study, which is to investigate the phosphorus solubilization ability of Priestia aryabhattai under the stress of Cd. This gives readers a clear understanding of what to expect in the paper. The introduction provides a good background on the topic, explaining the role of phosphate solubilizing bacteria in remediating Cd-contaminated soil and the potential of Priestia aryabhattai in this context. It includes references to previous studies, which helps to establish the context and significance of the research.

The flow of information could be improved. The introduction seems to jump between different topics, which might confuse readers. It would be beneficial to organize the information in a more logical and coherent manner. While the objective of the study is clear, the introduction does not sufficiently state the hypothesis or expected outcomes of the research. Including this could help to guide the reader's understanding of the study.

Overall, the introduction provides a good foundation for your research. With some improvements in language, flow, and clarity of hypothesis, it could be even stronger.

The document provides a comprehensive and detailed procedure for the experiment, including the preparation of the Priestia aryabhattai, the soil, and the culture medium. This level of detail enhances the reproducibility of the study. However, the document does not provide much background information on Priestia aryabhattai, such as its characteristics and why it was chosen for this study. This could limit the understanding of the study for those not familiar with this organism.

Also there are outlined rigorous measurement methods for determining soil physical and chemical properties, including the use of a fully automated Kjeldahl nitrogen analyzer, UV-Vis spectrophotometer, and Inductively Coupled Plasma Mass Spectrometer (ICP-MS). These methods ensure accurate and reliable results. However, since the soil was left to equilibrate and age for 90 days after being mixed with the Cd solution, it's unclear why this specific time period was chosen, and whether it could affect the results.

While the document mentions that the soil was obtained from a farmland in Yuci District, Jinzhong, Shanxi, China, the information on the soil's initial contamination levels or other characteristics that could influence the results are not very consistent.

The study includes multiple treatments and replicates, which can help to ensure the robustness and reliability of the results. However, it is not clear mentioned the use of a control group in the experiment. Without a control group, it may be difficult to determine the true effects of the treatments.

The use of One-Way ANOVA for data analysis and the significance level set at P<0.05 ensures the statistical validity of the results.

The document mentions that the results were processed with Origin 2021, but does not provide details on what this processing involved. This could limit the reproducibility of the study.

The manuscript provides a meticulous explanation of the results and their implications, which enhances the understanding of the reader. It includes Figures and Tables to present the data, which makes it easier for readers to understand the results. The use of error bars in the figures also provides a visual representation of the variability in the data.

However, the manuscript could benefit from more context or background information to help the reader understand the significance of the findings. For example, it's not clear why the study focuses on Cd-contaminated soil and the role of Priestia aryabhattai. While the manuscript discusses correlations, it could benefit from a clearer explanation of what these correlations mean and their implications for the study.

The references are relatively new, mainly from the last 5 years and support the authors in the analysis of the existing situation.

There are some particular issues that should be clarified, such as:

R33: What should mean “with the excess rate as high as 7.0%” when referring to Cadmium toxicity?

R: 36-38: The authors affirm that: “The farmland soil was identified as safe when the concentration of Cd is between 0.6 and 4 mg kg-1 which is considered as the risk screening value and the risk control value”, but does not provide as a reference a document of an authority that establishes this limit.

Comments on the Quality of English Language

The authors should check again the manuscript and correct some small typos

Author Response

Thank you very much for taking the time to review this manuscript. We have made correction according to the comments point by point, which we hope meet with approval. Revised portion are marked in red in the paper and the responds to the reviewers

Comments 1:

The introduction seems to jump between different topics, which might confuse readers. It would be beneficial to organize the information in a more logical and coherent manner.

Response 1:

Thank you for your careful review of our manuscript. We have improved the logic and language expression of the introduction as follows:

The application of phosphorus fertilizers is consistently at a high level in farmland due to its low seasonal availability [6-8]. However, the frequent and large-scale application of phosphorus fertilizers can cause the accumulation of Cd in agricultural soils, leading to the environmental problem [9]. Therefore, increasing the utilization of soil P to reduce the input of low-quality phosphorus fertilizers is also an effective strategy to remediate Cd-contaminated farmland [10,11]. (see page 2, lines 46-50)

The interaction between phosphorus and cadmium elements is an effective means for phosphorus solubilizing bacteria to remediate Cd-contaminated farmland [20]. (see page 2, lines 66-67)

References

[6] Bilal, S.; Hazafa, A.; Ashraf, I.; Alamri, S.; Siddiqui, M.H.; Ramzan, A.; Qamar, N.; Sher, F.; Naeem, M. Biochemical and Molecular Responses Underlying the Contrasting Phosphorus Use Efficiency in Ryegrass Cultivars. Plants. 2023, 12, 1224. [https://doi.org/10.3390/plants12061224]

[7] Ma, H.M.; Yu, X.; Yu, Q.; Wu, H.H.; Zhang, H.L.; Pang, J.Y.; Gao, Y.Z. Maize/alfalfa intercropping enhances yield and phosphorus acquisition. Field Crops Research. 2023, 303, 109136. [https://doi.org/10.1016/j.fcr.2023.109136]

[8] Su, N.; Xie, G.X.; Mao, Z.W.; Li, Q.R.; Chang, T.; Zhang, Y.P.; Peng, J.W.; Rong, X.M.;Luo, G.W. The effectiveness of eight-years phosphorus reducing inputs on double cropping paddy: Insights into productivity and soil-plant phosphorus tradeoff. Science of the Total Environment. 2023, 866, 161429. [https://doi.org/10.1016/j.scitotenv.2023.161429]

[9] Zhang, T.R.; Li, T.; Zhou, Z.J.; Li, Z.Q.; Zhang, S.R.; Wang, G.Y.; Xu, X.X.; Pu, Y.L.; Jia, Y.X.; Liu, X.J. Cadmium-resistant phosphate-solubilizing bacteria immobilized on phosphoric acid-ball milling modified biochar enhances soil cadmium passivation and phosphorus bioavailability. Science of the total environment. 2023, 877, 162812. [http://dx.doi.org/10.1016/j.scitotenv.2023.162812]

[10] Duan, X.Y.; Zou, C.L.; Jiang, Y.F.; Yu, X.J. Effects of Reduced Phosphate Fertilizer and Increased Trichoderma Application on the Growth, Yield, and Quality of Pepper. Plants. 2023, 12, 2998. [https://doi.org/10.3390/plants12162998]

[11] Qi, W.Y.; Chen, H.; Wang, Z.; Xing, S.F.; Song, C.; Yan, Z.; Wang, S.G. Biochar-immobilized Bacillus megaterium enhances Cd immobilization in soil and promotes Brassica chinensis growth. Journal of Hazardous Materials. 2023, 458,131921. [https://doi.org/10.1016/j.jhazmat.2023.131921]

[20] Qin, S.M.; Zhang, H.Y.; He, Y.H; Chen, Z.J.; Yao, L.Y.; Han, H. Improving radish phosphorus utilization efficiency and inhibiting Cd and Pb uptake by using heavy metal-immobilizing and phosphatesolubilizing bacteria. Science of the Total Environment. 2023, 868, 161685. [https://doi.org/10.1016/j.scitotenv.2023.161685]

Comments 2:

While the objective of the study is clear, the introduction does not sufficiently state the hypothesis or expected outcomes of the research. Including this could help to guide the reader's understanding of the study.

Response 2:

Thank you for your careful review of our manuscript. We have supplemented the hypothesis of the research as follows:

In this study, we hypothesize that Priestia aryabhattai can influence phosphorus-cadmium interactions by increasing the bioavailability of phosphorus in the soil. To test our hypothesis, a strain of phosphate-solubilizing bacteria was isolated from agricultural soil and identified as Priestia aryabhattai in this study. The screened and domesticated Priestia aryabhattai was inoculated in the farmland soil with the light, moderate, and heavy Cd contamination. The specific objectives of this study were to analyze the phosphorus solubility of Priestia aryabhattai and the interaction with P and Cd and crops by determining the content of soil inorganic and organic P, the content of available Cd, as well as the changes of bacterial abundance and community structure. The results support the innovative idea that developing and applying plant-Priestia aryabhattai combined technology for Eco-remediating Cd-contaminated farmland and increasing the utilization efficiency of phosphorus. (see page 3, lines 87-97)

Comments 3:

The document does not provide much background information on Priestia aryabhattai, such as its characteristics and why it was chosen for this study. This could limit the understanding of the study for those not familiar with this organism.

Response 3:

Thank you for your useful suggests of our manuscript. Priestia aryabhattai selected in this study is a new type of phosphorus solubilizing bacteria, and is revealed as a powerful multi-stress-tolerant crop growth promoter. PSB have been proposed as a potentially effective strategy to simultaneously remediate P-barren and Cd-contaminated farmland. However, it remains unknown whether Priestia aryabhattai could be utilized as an efficient inoculant for Cd-contaminated farmland. Therefore, this study aimed to provide a new idea for the phytomicrobial remediation of cadmium-contaminated farmland by studying the interaction of Priestia aryabhattai on phosphorus and cadmium in soil. We have supplemented the relevant background of Priestia aryabhattai and the reasons for choosing Priestia aryabhattai in this study as follows:

Previous studies have documented Priestia aryabhattai is a powerful multi-stress-tolerant crop growth promoter, and can improve plant productivity and enhance defense mechanisms against biotic and abiotic stresses [15-16].(see page 2, lines 55-57)

However, The effect mechanisms of Priestia aryabhattai on the phosphorus fractions in the Cd-contaminated farmland are yet to be revealed. (see page 2, lines 64-65)

References

[15]Wang, Y.J.; Wang, Y.; Zhang, Q.; Fan, H.Z.; Wang, X.Y.; Wang, J.A.; Zhou, Y.; Chen, Z.Y.;Sun, F.J.; Cui, X.Y. Saline-Alkali Soil Property Improved by the Synergistic Effects of Priestia aryabhattai JL-5, Staphylococcus pseudoxylosus XW-4, Leymus chinensis and Soil Microbiota. International Journal of Molecular Sciences. 2023, 24, 7737. [https://doi.org/10.3390/ijms24097737]

[16]Shen, F.T.; Yen, J.H.; Liao, C.S.; Chen, W.C.; Chao, Y.T. Screening of Rice Endophytic Biofertilizers with Fungicide Tolerance and Plant Growth-Promoting Characteristics. Sustainability. 2019, 11, 1133. [https://doi.org/10.3390/su11041133]

Comments 4:

However, since the soil was left to equilibrate and age for 90 days after being mixed with the Cd solution, it's unclear why this specific time period was chosen, and whether it could affect the results.

Response 4:

Thank you for your careful review of our manuscript. Existing studies and related literature have shown that balanced aging for 90 days can make the different forms of heavy metals in soil reach a relatively stable state. We have supplemented the relevant references in the manuscript [30]. (see page3, line 111)

Reference

Teng, Z.D.; Zhao, X.; Yuan, J.J. Phosphate functionalized iron based nanomaterials coupled with phosphate solubilizing bacteria as an efficient remediation system to enhance lead passivation in soil. Journal of Hazardous Materials. 2021, 419, 126433.[https://doi.org/10.1016/j.jhazmat.2021.126433]

Wang, G.T.; Zhao, X.; Luo, W.Q; et al. Noval porous phosphate-solubilizing bacteria beads loaded with BC/nZVI enhanced the transformation of lead fractions and its microecological regulation mechanism in soil. Journal of Hazardous Materials. 2022, 437, 129402.  [https://doi.org/10.1016/j.jhazmat.2022.129402]

Luo, J.P.; Liao, G.C.; Banerjee S.; et al. Long-term organic fertilization promotes the resilience of soil multifunctionality driven by bacterial communities. Soil Biology and Biochemistry. 2023, 177, 108922. [https://doi.org/10.1016/j.soilbio.2022.108922]

Comments 5:

The information on the soil's initial contamination levels or other characteristics that could influence the results are not very consistent.

Response 5:

Thank you for your careful review of our manuscript. The focus of this study is on the effect of Bacillus aspergii on the interaction of phosphorus and cadmium. Inoculated bacteria and uninoculated bacteria were used as control variables. The initial soil cadmium background value was lower than the screening value of safe farmland. Therefore, it can be used as a control group for safe use of cadmium pollution concentration.

Comments 6:

It is not clear mentioned the use of a control group in the experiment. Without a control group, it may be difficult to determine the true effects of the treatments.

Response 6:

Thank you for your careful review of our manuscript. We have supplemented the information of the control group in this study as follows:

Three treatments were included: inoculation of Priestia aryabhattai (N-PSB), soil sterilization (S), and inoculation of Priestia aryabhattai after soil sterilization (S-PSB). The unpasteurized soil served as the control group (N). (see page 4, lines 128-130)

Comments 7:

The document mentions that the results were processed with Origin 2021, but does not provide details on what this processing involved. This could limit the reproducibility of the study.

Response 7:

Thank you for your careful review of our manuscript. All experiments were carried out in triplicate. Data were analyzed using SPSS 24.0 and graphs were prepared using Origin 2021. Specific modifications are as follows:

The graphs were prepared using Origin 2021. (see page 6, lines 180-181)

Comments 8:

It's not clear why the study focuses on Cd-contaminated soil and the role of Priestia aryabhattai. While the manuscript discusses correlations, it could benefit from a clearer explanation of what these correlations mean and their implications for the study.

Response 8:

Thank you for your careful review of our manuscript. In the introduction part of this study, the reasons for studying the role of Priestia aryabhattai in cadmium-contaminated soil have been explained. Specific reasons are as follows:

Priestia aryabhattai is a new type of phosphorus solubilizing bacteria, and is revealed as a powerful multi-stress-tolerant crop growth promoter. PSB have been proposed as a potentially effective strategy to simultaneously remediate P-barren and Cd-contaminated farmland. However, it remains unknown whether Priestia aryabhattai could be utilized as an efficient inoculant for Cd-contaminated farmland. Therefore, this study aimed to provide a new idea for the phytomicrobial remediation of cadmium-contaminated farmland by studying the interaction of Priestia aryabhattai on phosphorus and cadmium in soil. (see page 2, lines 55-57; and page 2, lines 64-65)

We provide explanations of the correlation analysis in the Correlation analysis section, and add the implications of this result for the study below:

Great positive correlation also exists between Ca8-P, Ca10-P, Al-P and Fe-P (P < 0.01), indicating that Priestia aryabhattai increased the content of available P mainly by facilitating the transformation of inorganic P forms (except O-P) to Ca2-P, and then increasing the content of available P. (see page 13, lines 332-335)

Priestia aryabhattai facilitated the conversion of stabilized P forms to active Ca2-P and LOP, and released more PO43- to inhibit Cd migration. (see page 13, lines 343-345)

Therefore, Priestia aryabhattai is expected to be a remediation agent for Cd-contaminated soil. (see page 14, lines 347-348)

Comments 9:

What should mean “with the excess rate as high as 7.0%” when referring to Cadmium toxicity?

Response 9:

Thank you for your careful review of our manuscript. 7% refers to the pollution detection point overstandard rate of Cd in China’s farmland. we have revised as follows:

Cadmium is one of the most biotoxic heavy metals, and the pollution detection point overstandard rate of Cd is 7.0% in China’s farmland, which is at the top of the eight excess heavy metals. (see page1, lines 33-34)

Comments 10:

The authors affirm that: “The farmland soil was identified as safe when the concentration of Cd is between 0.6 and 4 mg kg-1 which is considered as the risk screening value and the risk control value”, but does not provide as a reference a document of an authority that establishes this limit.

Response 10:

Thank you for your helpful suggestion. The authoritative document of the limit value in this study comes from the 《Risk control standard for soil contamination of agricultural land》(GB15618-2018) issued by the Ministry of Ecology and Environmentand State Administration for Market Regulation of China. We have supplemented the corresponding reference documents as follows:

The farmland soil was identified as safe when the concentration of Cd is between 0.6 and 4 mg kg-1 which is considered as the risk screening value and the risk control value (GB15618-2018). (see page 1, lines 37-39)

Reviewer 4 Report

Comments and Suggestions for Authors

An interesting work.

Title way too long. Hard to read

  In summary..." activity of phosphate solubilizing bacteria is easily inhibited by the toxicity of Cd."... Who says this?

  The keyword part still needs to be worked on

Introduction

L 37... the concentration of Cd is between 0.6 and 4 mg kg-1 which is considered as the risk screening 37 value and the risk control value... the bibliographic source is missing

  What is the optimal amount of phosphorus in the soil? It does not depend on the type of culture?...to add the values and the bibliographic sources

Regarding the theory part of the introduction, it should be revised and improved, adding some bibliographic sources for each idea developed

At the end of the introduction, the purpose of this study and the degree of novelty must be emphasized

  Materials and methods

  Materials and methods lack data regarding the apparatus/equipment used and the methods according to which the determinations were made in subsections 2.1-2.3

Was the soil sampled before it was polluted with CD investigated to see if there were any concentrations of metals in it? If so, what are they? What about the pH of the soil and the physico-chemical analyzes carried out in the usual way? Are the data in table 2 the initial data of the soil?

  In the results part, there are many tables... some of them can be made in graphic form so that the obtained results are as obvious as possible, having the possibility to compare as well as possible

The results should be compared with other studies, and to highlight the contribution of this study

In the part of the conclusion, one can insist on the basis of the obtained results and then on what is to be researched

Author Response

Thank you very much for taking time out of your busy schedule to give us constructive suggestions, which are very helpful for us to improve our paper. Please find the detailed responses below and the corresponding revisions/corrections highlighted/in track changes in the re-submitted files.

Comments 1:

Title way too long. Hard to read.

Response 1:

Thank you for your helpful suggestion. We have changed the title to make it easier for readers to read and understand as follow:

Effects of Priestia aryabhattai on Phosphorus Fraction and Implifications for Plant-microbe technology Eco-remediating Cd-contaminated farmland

Comments 2:

In summary..." activity of phosphate solubilizing bacteria is easily inhibited by the toxicity of Cd."... Who says this?

Response 2:

Thank you for your careful review of our manuscript. Previous studies have found that even at very low Cd concentrations, the associated toxicity could lead to low Phosphate-solubilizing bacteria activity and survival during bioremediation (Rahman, 2020; Zhang et al., 2023). Therefore, it is mentioned in the abstract that the activity of phosphate solubilizing bacteria is easily inhibited by the toxicity of Cd.

Reference

Rahman, Z. An overview on heavy metal resistant microorganisms for simultaneous treatment of multiple chemical pollutants at co-contaminated sites, and their multipur-pose application. Journal of Hazardous Materials. 2020, 396, 122682. [https://doi.org/10.1016/j.jhazmat.2020.122682]

Zhang, T.R.; Li, T.; Zhou, Z.J.; Li, Z.Q.; Zhang, S.R.; Wang, G.Y.; Xu, X.X.; Pu, Y.L.; Jia, Y.X.; Liu, X.J. Cadmium-resistant phosphate-solubilizing bacteria immobilized on phosphoric acid-ball milling modified biochar enhances soil cadmium passivation and phosphorus bioavailability. Science of the total environment. 2023, 877, 162812. [http://dx.doi.org/10.1016/j.scitotenv.2023.162812]

Comments 3:

The keyword part still needs to be worked on.

Response 3:

Thank you for your careful review of our manuscript. In the key words, we unified the expressions of phosphorus fractions and Cd-contaminated farmland, as well as the font format of Priestia aryabhattai as follows:

Keywords: Plant-microbe combined technology, Soil remediation, Phosphorus fractions, Priestia aryabhattai, Physio-chemical-micrbial mechanism, Eco-remediating Cd-contaminated farmland (see page 1, lines 29-30)

Comments 4:

The concentration of Cd is between 0.6 and 4 mg kg-1 which is considered as the risk screening value and the risk control value... the bibliographic source is missing

Response 4:

Thank you for your careful review of our manuscript. We have supplemented the relevant standard documents as follows:

The farmland soil was identified as safe when the concentration of Cd is between 0.6 and 4 mg kg-1 which is considered as the risk screening value and the risk control value (GB 15618-2018). (see page 1, lines 38-39)

Comments 5:

What is the optimal amount of phosphorus in the soil? It does not depend on the type of culture?...to add the values and the bibliographic sources

Response 5:

Thank you for your careful review of our manuscript. The total phosphorus content of large loess calcareous soils in China is approximately 0.57 to 0.87g kg-1, but due to a large amount of free calcium carbonate, most of the phosphorus becomes insoluble calcium phosphate, and the total phosphorus content of the soil in this study is within this range.

Reference

Bao, S.D. Soil and Agricultural Chemistry Analysis. Chinese Agricultural Press. Beijing (In Chinese), 2000.

Comments 6:

Regarding the theory part of the introduction, it should be revised and improved, adding some bibliographic sources for each idea developed

Response 6:

Thank you for your helpful suggestion. We have improved the logic and language expression of the introduction as follows:

The application of phosphorus fertilizers is consistently at a high level in farmland due to its low seasonal availability [6-8]. However, the frequent and large-scale application of phosphorus fertilizers can cause the accumulation of Cd in agricultural soils, leading to the environmental problem [9]. Therefore, increasing the utilization of soil P to reduce the input of low-quality phosphorus fertilizers is also an effective strategy to remediate Cd-contaminated farmland [10,11]. (see page 2, lines 46-50)

The interaction between phosphorus and cadmium elements is an effective means for phosphorus solubilizing bacteria to remediate Cd-contaminated farmland [20]. (see page 2, lines 66-67)

References

[6] Bilal, S.; Hazafa, A.; Ashraf, I.; Alamri, S.; Siddiqui, M.H.; Ramzan, A.; Qamar, N.; Sher, F.; Naeem, M. Biochemical and Molecular Responses Underlying the Contrasting Phosphorus Use Efficiency in Ryegrass Cultivars. Plants. 2023, 12, 1224. [https://doi.org/10.3390/plants12061224]

[7] Ma, H.M.; Yu, X.; Yu, Q.; Wu, H.H.; Zhang, H.L.; Pang, J.Y.; Gao, Y.Z. Maize/alfalfa intercropping enhances yield and phosphorus acquisition. Field Crops Research. 2023, 303, 109136. [https://doi.org/10.1016/j.fcr.2023.109136]

[8] Su, N.; Xie, G.X.; Mao, Z.W.; Li, Q.R.; Chang, T.; Zhang, Y.P.; Peng, J.W.; Rong, X.M.;Luo, G.W. The effectiveness of eight-years phosphorus reducing inputs on double cropping paddy: Insights into productivity and soil-plant phosphorus tradeoff. Science of the Total Environment. 2023, 866, 161429. [https://doi.org/10.1016/j.scitotenv.2023.161429]

[9] Zhang, T.R.; Li, T.; Zhou, Z.J.; Li, Z.Q.; Zhang, S.R.; Wang, G.Y.; Xu, X.X.; Pu, Y.L.; Jia, Y.X.; Liu, X.J. Cadmium-resistant phosphate-solubilizing bacteria immobilized on phosphoric acid-ball milling modified biochar enhances soil cadmium passivation and phosphorus bioavailability. Science of the total environment. 2023, 877, 162812. [http://dx.doi.org/10.1016/j.scitotenv.2023.162812]

[10] Duan, X.Y.; Zou, C.L.; Jiang, Y.F.; Yu, X.J. Effects of Reduced Phosphate Fertilizer and Increased Trichoderma Application on the Growth, Yield, and Quality of Pepper. Plants. 2023, 12, 2998. [https://doi.org/10.3390/plants12162998]

[11] Qi, W.Y.; Chen, H.; Wang, Z.; Xing, S.F.; Song, C.; Yan, Z.; Wang, S.G. Biochar-immobilized Bacillus megaterium enhances Cd immobilization in soil and promotes Brassica chinensis growth. Journal of Hazardous Materials. 2023, 458,131921. [https://doi.org/10.1016/j.jhazmat.2023.131921]

[20] Qin, S.M.; Zhang, H.Y.; He, Y.H; Chen, Z.J.; Yao, L.Y.; Han, H. Improving radish phosphorus utilization efficiency and inhibiting Cd and Pb uptake by using heavy metal-immobilizing and phosphatesolubilizing bacteria. Science of the Total Environment. 2023, 868, 161685. [https://doi.org/10.1016/j.scitotenv.2023.161685]

Comments 7:

At the end of the introduction, the purpose of this study and the degree of novelty must be emphasized.

Response 7:

Thank you for your careful review of our manuscript. We have supplemented the purpose of this study and the degree of novelty as follows:

In this study, we hypothesize that Priestia aryabhattai can influence phosphorus-cadmium interactions by increasing the bioavailability of phosphorus in the soil. To test our hypothesis, a strain of phosphate-solubilizing bacteria was isolated from agricultural soil and identified as Priestia aryabhattai in this study. The screened and domesticated Priestia aryabhattai was inoculated in the farmland soil with the light, moderate, and heavy Cd contamination. The specific objectives of this study were to analyze the phosphorus solubility of Priestia aryabhattai and the interaction with P and Cd and crops by determining the content of soil inorganic and organic P, the content of available Cd, as well as the changes of bacterial abundance and community structure. The results support the innovative idea that developing and applying plant-Priestia aryabhattai combined technology for Eco-remediating Cd-contaminated farmland and increasing the utilization efficiency of phosphorus. (see page 3, lines 87-97)

Comments 8:

Materials and methods lack data regarding the apparatus/equipment used and the methods according to which the determinations were made in subsections 2.1-2.3

Response 8:

Thank you for your helpful suggestion. We have supplemented the relevant references to the experimental methods in parts 2.1-2.3 of the manuscript as follows:

The soil was well-mixed with each Cd solution, then left to equilibrate and age for 90 days [30]. (see page 3, line 111)

The agar medium was made from 10.0 g of C6H12O6, 5.0 g of Ca3(PO4)2 or Hydroxyapatite, 5.0 g of MgCl2·6H2O, 0.25 g of MgSO4·7H2O, 0.20 g of KCl, 0.10 g of (NH4)2SO4 and 18 g-20 g of agar [31]. (see page 3, line 115)

Comments 9:

Was the soil sampled before it was polluted with Cd investigated to see if there were any concentrations of metals in it? If so, what are they? What about the pH of the soil and the physico-chemical analyzes carried out in the usual way? Are the data in table 2 the initial data of the soil?

Response 9:

Thank you for your careful review of our manuscript. The initial contents of total Cd (0.056±0.003 mg kg-1) and available Cd (0.027±0.003 mg kg-1) before cadmium pollution as well as the soil physical and chemical properties were determined, as shown in Table 2.

Comments 10:

In the results part, there are many tables... some of them can be made in graphic form so that the obtained results are as obvious as possible, having the possibility to compare as well as possible.

Response 10:

Thank you for your careful review of our manuscript. There are many experimental treatment groups involved in this study, including the treatment of cadmium with different concentrations and the treatment of soil inoculated bacteria/non-inoculated bacteria. Considering the large amount of data, in order to facilitate readers to obtain data information more conveniently, we choose to present the data in the form of tables.

Comments 11:

The results should be compared with other studies, and to highlight the contribution of this study

Response 11:

Thank you for your helpful suggestion. We have added a comparison of the results with other studies as follows:

As shown in Table 7, the treatment of N4-PSB with the application of Priestia aryabhattai slightly decreased Chao1, Pielou evenness, Shannon and Simpson as compared with the treatment of N4, which was consistent with the research results of Zhang et al [9]. This may be attributed to the competition between indigenous and inoculated bacteria [53]. (see page 14, lines 362-364)

In the conclusion part, we explain the significance and contribution of this study as follows:

This sudy firstly provides systematic chemical mechanism for plant-microbe combined technology Eco-remediating Cd-contaminated farmland. Focusing on the study of phosphorus fertilizers in cadmium-contaminated farmland, the transformation and availability of phosphorus fractions in soil under the application of PSB were analyzed in detail, which provided solid theoretical foundation for the remediation of Cd-contaminated farmland by plant-microbe combined technology, and also presented a new solution to increase the seasonal availability of phosphorus fertilizer. (see page 19, lines 438-444)

Reference

Zhang, T.R.; Li, T.; Zhou, Z.J.; Li, Z.Q.; Zhang, S.R.; Wang, G.Y.; Xu, X.X.; Pu, Y.L.; Jia, Y.X.; Liu, X.J. Cadmium-resistant phosphate-solubilizing bacteria immobilized on phosphoric acid-ball milling modified biochar enhances soil cadmium passivation and phosphorus bioavailability. Science of the total environment. 2023, 877, 162812. [http://dx.doi.org/10.1016/j.scitotenv.2023.162812]

Qi, X.; Xiao, S.; Chen, X.; Ali, I.; Gou, J.L.; Wang, D.; Zhu, B.; Zhu, W.K.;Shang, R.; Han, M.W. Biochar-based microbial agent reduces U and Cd accumulation in vegetables and improves rhizosphere microecology. Journal of Hazardous Materials. 2022, 436, 129147. [https://doi.org/10.1016/j.jhazmat.2022.129147]
